# Physiological and Transcriptomic Mechanisms Underlying Vitamin C-Mediated Cold Stress Tolerance in Grafted Cucumber

**DOI:** 10.3390/plants14152398

**Published:** 2025-08-02

**Authors:** Panpan Yu, Junkai Wang, Xuyang Zhang, Zhenglong Weng, Kaisen Huo, Qiuxia Yi, Chenxi Wu, Sunjeet Kumar, Hao Gao, Lin Fu, Yanli Chen, Guopeng Zhu

**Affiliations:** 1School of Breeding and Multiplication, Sanya Institute of Breeding and Multiplication, Hainan University, Sanya 572025, China; panpanyu0102@163.com (P.Y.); 20223007390@hainanu.edu.cn (J.W.); 18637562844@163.com (X.Z.); w2282072550@163.com (Z.W.); wsqx926@163.com (Q.Y.); chenxii77@163.com (C.W.); kumarsunjeet082@gmail.com (S.K.); gaohao1288@gmail.com (H.G.); 15672716152@163.com (L.F.); 2Key Laboratory for Quality Regulation of Tropical Horticultural Crops of Hainan Province, Tropical Agriculture and Forestry College, Hainan University, Haikou 570228, China; 3Institute of Tropical Bioscience and Biotechnology, Chinese Academy of Tropical Agricultural Sciences, Haikou 571101, China; kaisenbass@163.com

**Keywords:** grafted cucumber seedlings, low-temperature stress, vitamin C, transcriptome analysis

## Abstract

Cucumbers (*Cucumis sativus* L.) are highly sensitive to cold, but grafting onto cold-tolerant rootstocks can enhance their low-temperature resilience. This study investigates the physiological and molecular mechanisms by which exogenous vitamin C (Vc) mitigates cold stress in grafted cucumber seedlings. Using cucumber ‘Chiyu 505’ as the scion and pumpkin ‘Chuangfan No.1’ as the rootstock, seedlings were grafted using the whip grafting method. In the third true leaf expansion stage, seedlings were foliar sprayed with Vc at concentrations of 50, 100, 150, and 200 mg L^−1^. Three days after initial spraying, seedlings were subjected to cold stress (8 °C) for 3 days, with continued spraying. After that, morphological and physiological parameters were assessed. Results showed that 150 mg L^−1^ Vc treatment was most impactive, significantly reducing the cold damage index while increasing the root-to-shoot ratio, root vitality, chlorophyll content, and activities of antioxidant enzymes (SOD, POD, CAT). Moreover, this treatment enhanced levels of soluble sugars, soluble proteins, and proline compared to control. However, 200 mg L^−1^ treatment elevated malondialdehyde (MDA) content, indicating potential oxidative stress. For transcriptomic analysis, leaves from the 150 mg L^−1^ Vc and CK treatments were sampled at 0, 1, 2, and 3 days of cold stress. Differential gene expression revealed that genes associated with photosynthesis (*LHCA1*), stress signal transduction (*MYC2-1*, *MYC2-2*, *WRKY22*, *WRKY2*), and antioxidant defense (*SOD-1*, *SOD-2*) were initially up-regulated and subsequently down-regulated, as validated by qRT-PCR. Overall, we found that the application of 150 mg L^−1^ Vc enhanced cold tolerance in grafted cucumber seedlings by modulating gene expression networks related to photosynthesis, stress response, and the antioxidant defense system. This study provides a way for developing Vc biostimulants to enhance cold tolerance in grafted cucumbers, improving sustainable cultivation in low-temperature regions.

## 1. Introduction

Hainan Province serves as a critical production hub for winter vegetables supplied to northern China, with cucumber (*Cucumis sativus* L.) being a key crop. Over the past four decades, Hainan has frequently experienced cold winters, with temperatures dropping below 8 °C in regions north of Wuzhishan Mountain during winter and spring. These low temperatures pose a significant challenge to cucumber production due to cold sensitivity. Grafting, particularly whip grafting, is an impactive technique to enhance cold tolerance in cucumber seedlings [1,2]. Exogenous application of vitamin C (Vc) has shown promise in mitigating low-temperature stress by inhibiting lipid peroxidation, reducing malondialdehyde (MDA) accumulation, and enhancing antioxidant capacity, thereby protecting plants from cold-induced damage [3]. Vc supports the photosynthesis and antioxidant systems under cold stress, reduces reactive oxygen species (ROS) accumulation, and improves root system vitality [4]. Previous studies on tomato plants have demonstrated that exogenous Vc alleviated cold stress by regulating antioxidant functions, osmotic adjustments, ionic homeostasis, and the expression of catalase and heat shock proteins [5].

As an antioxidant and metabolic cofactor, Vc plays an important role in plant responses to abiotic stress by coordinating signaling pathways [6]. In longan fruits, exogenous Vc application has been shown to preserve sucrose and total soluble sugar contents, reducing weight loss, disease incidence, and skin browning [7]. Additionally, Vc mitigates cold-induced damage to the photosynthetic and antioxidant systems, alleviating excess light energy and photoinhibition in seedling leaves. Plants have evolved multiple mechanisms to cope with low-temperature stress, including the rapid induction of transcription factors (TFs) such as C-repeat binding factors (*CBFs*) and cold-regulated (*COR*) genes. *CBFs*, regulated by upstream TFs like ICE1, are critical for cold tolerance by modulating *COR* gene expression in the ICE-CBF-COR signaling pathway [8,9,10,11]. This pathway holds a significant potential for improving cold tolerance in cucurbit crops [12]. Transcriptomic analysis of cucumber and melon seedlings under cold stress has identified differentially expressed genes (DEGs) primarily involved in photosynthesis, carbon metabolism, and hormone signaling, with key TFs including *WRKY*, *AP2/ERF*, and *MYB*. Notably, *CBF* genes from the AP2 family respond rapidly to cold stress, while *WRKY* family members, such as *CsWRKY21* and *CsWRKY46*, positively regulate cold tolerance in Arabidopsis [13].

Despite the potential of Vc in alleviating cold stress in cucurbit crops, research has primarily focused on watermelon and melon, with limited systematic studies on grafted cucumber seedlings. The optimal Vc concentration, its physiological impacts, and the underlying molecular mechanisms in grafted cucumber seedlings under cold stress remain unexplored. Determining the optimum Vc concentration and elucidating its mechanisms are essential for optimizing cold-tolerance cultivation practices and ensuring stable winter yields in Hainan Province. This study investigates the impacts of different concentrations of Vc on cucumber ‘Chiyu 505’ grafted onto pumpkin ‘Chuangfan No.1’ rootstock using the whip grafting method at the third true leaf expansion stage under low-temperature stress. Morphological and physiological indicators were analyzed, and transcriptomic profiling was conducted to uncover the molecular mechanisms of cold tolerance, providing a theoretical basis for enhancing cucumber production during Hainan’s winter–spring season.

## 2. Results

### 2.1. Impact of Vc on Morphological Characteristics of Grafted Cucumber Seedlings Under Low-Temperature Stress

The morphological responses of grafted cucumber seedlings to different Vc concentrations under low-temperature stress are presented in Table 1. Among all treatments, A3 (150 mg L^−1^ Vc) demonstrated the most significant improvement in growth parameters. The stem diameter of scions in treatment A3 showed a marked increase compared to other treatments (*p* < 0.05). Compared with CK, the stem diameter of the scions increased significantly by 9.4%, the plant height increased significantly by 10.53%, the leaf area increased significantly by 34.41%, the fresh weight increased significantly by 25.14%, and the cold damage index decreased significantly by 10.69% (Table 1). This indicates that spraying an appropriate concentration of Vc can alleviate the inhibitory effect of low temperature on plant growth and, at the same time, can alleviate the stress of low temperature on plants and reduce the cold damage index of plants (Figure 1).

### 2.2. Impact of Vc on Physiological Indicators of Grafted Cucumber Seedlings Under Low-Temperature Stress

Low-temperature stress led to a significant decrease in the content of photosynthetic pigments in plants (Figure 2A). However, after Vc treatment, this negative impact was effectively reversed, resulting in a substantial increase in the content of photosynthetic pigments under salt stress conditions. Specifically, the chlorophyll content in the Vc treatment group was significantly higher than that in the CK group, and the A3 treatment increased by 15.21%. By preserving photosynthetic pigments, the Vc treatment helps to improve the overall performance of plant photosynthesis and energy capture, which is crucial for the ability of plants to tolerate and thrive under low-temperature stress.

Vc had a significant impact on the morphological characteristics of the root system, showing a notable improvement compared to CK. Among them, the A3 treatment increased the root vitality and root–shoot ratio by 16.41% and 11.76%, respectively (Figure 2B,C). These significant improvements in root traits indicate that Vc plays a crucial role in enhancing the stress resistance of plant roots.

Under different concentrations of vitamin C treatments, there was no significant difference in the growth index of the seedlings among all treatments, but the growth index of the A3 treatment was the highest (Figure 2D).

### 2.3. Impact of Vc on Permeating Substance Content of Grafted Cucumber Seedlings Under Low-Temperature Stress

Low-temperature stress combined with the application of Vc has a significant impact on the oxidative damage of plant cells. This is confirmed by the substantial increase in proline (pro) accumulation and the reduction in malondialdehyde (MDA) production. Compared to CK, the proline accumulation under Vc treatment was significantly increased, and the MDA content was significantly decreased. However, both showed a trend of first decreasing and then increasing. Among them, the proline content under A2 treatment was significantly increased by 28.19% compared to CK, while A4 treatment only increased by 2.45% compared to CK (Figure 3A); the MDA content was significantly reduced by 14.85% compared to CK, and A4 treatment increased by 5.8% compared to CK (Figure 3B). This indicates that a low concentration of Vc helps to alleviate the oxidative damage caused by low-temperature stress to plants, while a high concentration has the opposite effect.

### 2.4. Impact of Vc on the Content of Osmotic Regulatory Substances in Grafted Cucumber Seedlings Under Low-Temperature Stress

Under low-temperature stress, plants need osmotic regulatory substances to survive better. Under low-temperature stress conditions, Vc treatment significantly increased the content of osmotic regulatory substances. The contents of soluble proteins and soluble sugars showed a trend of increasing first and then decreasing, but both were significantly higher than CK. The soluble protein content of A3 treatment significantly increased by 19.1% (Figure 4A); among them, the soluble sugar content of A2 treatment was significantly increased by 30.23% compared to CK, and that of A4 treatment was only 15.11% higher than CK (Figure 4B). This indicates that Vc shows good effects in enhancing the plant’s ability to resist oxidative damage induced by salt through enhancing the osmotic regulatory substances.

### 2.5. Impact of Vc on Antioxidant Enzyme Activities in Grafted Cucumber Seedlings Under Low-Temperature Stress

Compared with CK, with the increase in Vc concentration, the activities of antioxidant enzymes in each treatment showed a trend of first increasing and then decreasing, and all were significantly higher than CK. The activities of all antioxidant enzymes in the A3 treatment were the highest. Compared with CK, the activities of SOD, POD, and CAT were significantly increased by 64.12%, 16.36%, and 36.27%, respectively (Figure 5A–C). The increase in antioxidant enzyme activity helps to reduce oxidative damage and enhances the plant’s resistance to low-temperature stress.

### 2.6. RNA Sequencing, Assembly, and Analysis of Differential Genes

The transcriptomic analysis of 24 samples generated 147.96 Gb of clean data, with each sample yielding ≥ 5.88 Gb and Q30 ≥ 95.28% (Appendix A). Reference genome alignment efficiency ranged between 93.49% and 98.95% across samples, confirming data suitability for detailed analysis (Figure 6A).

Differential expression analysis identified 3852 DEGs between CA and AA (2288 up-regulated), 2232 between CB and AB (1022 up-regulated), 733 between CC and AC (150 up-regulated), and 343 between CD and AD (198 up-regulated) (Figure 6B). Notably, only two similar genes showed differential expression across all comparisons, with CA and AA containing the majority of unique DEGs (2901) (Figure 6C).

### 2.7. GO Functional Enrichment and KEGG Pathway Annotation Analysis of DEGs

Gene Ontology (GO) enrichment analysis categorized the DEGs into three functional domains (Figure 7). Biological processes were primarily associated with cellular and metabolic processes and biological regulation. Cellular components showed enrichment in cellular anatomical entities, intracellular structures, and protein complexes. Molecular functions were dominated by binding activity, catalytic activity, and transporter activity.

KEGG (Kyoto Encyclopedia of Genes and Genomes) pathway analysis revealed significant enrichment in five major categories, with the most represented pathways being plant–pathogen interaction, plant hormone signal transduction, MAPK signaling pathway–plant, starch and sucrose metabolism, and carbon metabolism (Figure 8).

### 2.8. Key Genes in Signal Transduction Pathways Under Low-Temperature Stress

Focusing on cold stress resistance differences between CK and A3, which showed abundant DEGs and differentially accumulated metabolites (DAMs) (Figure 7 and Figure 8). Transcriptomic analysis indicated that DEGs in AD treatment were enriched in metabolic pathways, including α-linolenic acid metabolism, phenylpropanoid biosynthesis, and carbohydrate metabolism, with carbohydrate metabolism being the most enriched. Key pathways, such as starch and sucrose metabolism and glycolysis/gluconeogenesis, were significantly enriched in both CD and AD treatments. Genes related to these pathways, including fructose-bisphosphate aldolase (ALOD), sucrose synthase (SUS), diphosphate-dependent phosphofructokinase (PFP), and 2,3-bisphosphoglycerate-dependent phosphoglycerate mutase (PGAM), were predominantly down-regulated in CD treatment but up-regulated in AD (Figure 9). These findings suggest that Vc at 150 mg L^−1^ modulates carbohydrate metabolism to enhance cold stress tolerance in grafted cucumber seedlings.

### 2.9. qRT–PCR Verification

To validate the transcriptome data, we selected nine representative DEGs for qRT-PCR (Figure 10). The expression patterns observed in RNA-seq of these genes (*LHCA1*, *MYC2-1*, *MYC2-2*, *WRKY22*, *WRKY2*, *SOD-1*, and *SOD-2*) showed initial up-regulation followed by down-regulation, which were confirmed by qRT-PCR. In contrast, *PAL* exhibited consistent down-regulation, while the *carbonic anhydrase* gene displayed down-regulation followed by up-regulation. The qRT-PCR results strongly correlated with transcriptome data (R^2^ = 0.7546, Appendix A), confirming the reliability of our RNA-seq findings. These consistent findings verify the gene expression changes detected under low-temperature stress.

## 3. Discussion

Grafting enhances cold tolerance in cucumber seedlings by leveraging the strong root system of cold-tolerant rootstocks, such as pumpkin ‘Chuangfan No.1’, to improve cell membrane stability, osmotic adjustment, and antioxidant enzyme activity [14]. Vitamin C (Vc), a key antioxidant in plant tissues, mitigates oxidative stress induced by low temperatures, which triggers excessive ROS production, disrupting redox balance and causing oxidative damage [15]. Antioxidant enzymes, including SOD, POD, and CAT, counteract ROS to maintain cellular integrity [16]. This study indicates that exogenous Vc, particularly at 150 mg L^−1^, significantly improves cold tolerance in cucumber seedlings grafted onto rootstock using the whip grafting method at the third true leaf expansion stage under 8 °C stress.

Physiological results showed that Vc at 150 mg L^−1^ (treatment A3) optimized seedling vigor, with the highest root-to-shoot ratio (0.076), root activity (300.5 μg g^−1^ h^−1^), and chlorophyll content (38.4 mg g^−1^) compared to other treatments and control (CK). These improvements reflect enhanced biomass accumulation, photosynthetic capacity, and root development, reducing the cold injury index and promoting growth under stress. Ability of Vc to inhibit lipid peroxidation was evident, as treatment 3 exhibited the lowest MDA content (18.64 nmol g^−1^), indicating reduced membrane damage [17]. However, A4 (200 mg L^−1^) showed the highest MDA content, suggesting that excessive Vc may act as a pro-oxidant, exacerbating cellular damage. This highlights the importance of precise Vc dosing to maximize antioxidant benefits.

Osmotic adjustment is crucial for cold stress adaptation, with soluble sugars, soluble proteins, and proline serving as osmolytes to maintain cellular water balance and stabilize membranes [18]. Treatments A2 (100 mg L^−1^) and A3 (150 mg L^−1^) significantly increased soluble sugars, proteins, and proline contents compared to CK, with A3 showing optimal effects. These results indicate that Vc promotes osmolyte biosynthesis, lowering water potential, enhancing water retention, and supporting protein stability under cold stress [18], consistent with findings in other crops like longan [7].

Antioxidant enzyme activities were markedly enhanced by Vc, with treatment A3 showing the highest SOD (289.98 U g^−1^), POD (329.17 U g^−1^ min^−1^), and CAT (174.56 U g^−1^ min^−1^) activities. These enzymes efficiently scavenge ROS, preserving membrane integrity and mitigating cold-induced damage [16]. The dose-dependent response, peaking at 150 mg L^−1^ and declining at 200 mg L^−1^, highlights the optimal Vc concentrations for antioxidant defense. Collectively, these physiological enhancements confirm that 150 mg L^−1^ Vc is most effective for alleviating cold stress in grafted cucumber seedlings.

Low-temperature stress triggers plant membrane solidification, causing cell damage, enzyme disruption, ROS accumulation, and growth retardation, potentially leading to cell death [19,20]. Therefore, this experiment analyzed the role of Vc in alleviating low-temperature effects in grafted seedlings and explored its molecular mechanism.

Transcriptomic analysis elucidated the molecular mechanisms of Vc-mediated cold tolerance [21]. High-quality RNA-seq data (147.96 Gb, Q30 ≥ 95.28%, alignment efficiency (93.49–98.95%)) identified 343 DEGs in treatment A3 (AD) vs. CK (CD) at day 3 of cold stress, with 198 up-regulated and 145 down-regulated genes. GO enrichment showed DEGs associated with cellular processes, metabolic processes, biological regulation, binding, catalytic activity, and transporter activity [22]. KEGG analysis revealed enrichment in plant–pathogen interaction, plant hormone signal transduction, MAPK signaling, starch and sucrose metabolism, and carbon metabolism, indicating that Vc regulates a broad stress-responsive network.

Key DEGs, including *LHCA1-4* (photosynthesis), *MYC2-1*, *MYC2-2*, *WRKY22*, *WRKY2* (stress signaling), and *SOD-1*, *SOD-2* (antioxidant defense), displayed initial up-regulation followed by down-regulation, reflecting a dynamic stress response. LHCA1 likely enhances photosynthetic efficiency [23,24], while *MYC2* genes mediate jasmonic acid signaling in mild cold stress [25]. WRKY TFs regulate abiotic stress responses [26], and SOD genes scavenge ROS under drought, salt, heat, and cold stress [27]. Up-regulation of starch and sucrose metabolism and glycolysis/gluconeogenesis genes, such as fructose-bisphosphate aldolase (ALOD), sucrose synthase (SUS), diphosphate-dependent phosphofructokinase (PFP), and 2,3-bisphosphoglycerate-dependent phosphoglycerate mutase (PGAM), in treatment A3 suggests energy metabolism for cold tolerance.

This study found that 150 mg L^−1^ Vc has the ability to enhance cold tolerance in grafted cucumber seedlings by improving physiological parameters and modulating gene expression in stress-responsive pathways. However, this study was limited to a single cultivar, a short cold stress duration, and leaf-specific transcriptomics. Future research could explore the impact on diverse cultivars, longer stress periods, in-depth genomic investigations, and field trials to optimize Vc application. This study lays a foundation for developing climate-resilient cucumber cultivation strategies, guiding future studies on antioxidant-based stress mitigation in horticulture.

## 4. Materials and Methods

### 4.1. Experimental Site and Materials

The experiment was conducted in the greenhouse of Hainan University, Haikou, China, from October 2023 to January 2024. The cucumber scion cultivar ‘Chiyu 505’ was obtained from Tianjin Kerun Agricultural Technology Co., Ltd. (Tianjin, China), and the pumpkin rootstock cultivar ‘Chuangfan No.1’ was sourced from Sanya Tengnong Technology Co., Ltd. (Sanya, China).

### 4.2. Grafting and Treatment Application

Cucumber seedlings were grafted onto pumpkin rootstocks using the whip grafting method and grown until the third true leaf expansion stage. Seedlings were then subjected to foliar spraying with Vc at concentrations of 50 mg L^−1^ (A1), 100 mg L^−1^ (A2), 150 mg L^−1^ (A3), and 200 mg L^−1^ (A4) and distilled water as the control (CK). Spraying was performed daily until the leaves were fully wetted, and this continued for three days. Afterward, seedlings were transferred to an incubator set at 8 °C with a photoperiod of 12/12 h (light/dark) for low-temperature treatment. Vc spraying continued during the three-day stress period. Each treatment comprised 50 plants with triplicates.

For molecular analysis, based on physiological results indicating 150 mg L^−1^ as the optimal Vc concentration, additional seedlings at the third true leaf expansion stage were sprayed with either 150 mg L^−1^ Vc (experimental group) or distilled water (control group) daily for 3 days. These seedlings were then subjected to 8 °C with a photoperiod of 12/12 h. Leaf samples were collected at 0, 1, 2, and 3 days of cold stress, with three biological replicates per time point. Control group samples were labelled CA, CB, CC, and CD (days 0, 1, 2, and 3, respectively), and experimental group samples were labeled AA, AB, AC, and AD (days 0, 1, 2, and 3, respectively). Samples were immediately frozen at −80 °C.

### 4.3. Measurement of Morphological and Physiological Indicators

At the end of the three-day cold stress period, three grafted seedlings per replicate were randomly selected for analysis. Plants were washed, and plant height and stem diameter were measured using a ruler and Vernier caliper, respectively. Fresh weights of the aboveground and underground parts were measured using an electronic balance with a precision of 0.001 g. Seedlings were then placed in an oven at 105 °C for 30 min to deactivate enzymes, followed by drying at 80 °C to constant weight to determine dry weights. Leaf area was calculated following the method of Pei [28]. A root scanner (Epson 11000 XL, Epson, Suwa, Japan) was used to measure root length, surface area, diameter, and volume of the grafted seedlings. The chilling injury index was determined using the method of Wang [29].

Seedling vigor was evaluated using the following indices:Root-to-shoot ratio = root dry weight/(rootstock stem dry weight + scion dry weight)Vigor index = scion stem diameter/scion height × whole plant dry weight

Root activity of grafted seedlings was measured using the 2,3,5-triphenyltetrazolium chloride (TTC) staining method. Chlorophyll content in grafted seedlings was measured using a portable chlorophyll meter (SPAD-502, Konica Minolta, Tokyo, Japan), averaging three measurements per true leaf. MDA content was measured using the thiobarbituric acid (TBA) method, and proline content was determined using the sulfosalicylic acid method [30]. Likewise, soluble sugar content was measured using the anthrone method, and soluble protein content was determined using the Coomassie Brilliant Blue staining method [31]. POD activity was measured using the guaiacol method [32,33], SOD activity was measured using the nitrogen blue tetrazolium (NBT) reduction method [34], and CAT activity was determined using the ultraviolet absorption method [35].

### 4.4. Transcriptome Sequencing and Differential Expression Gene Screening

Transcriptome sequencing of leaf samples was performed by Beijing Biomarker Technologies Co., Ltd. (Beijing, China). The RNA-seq sequencing experimental process includes sample detection, library preparation, and quality control, as well as sequencing. Bioinformatics analysis was conducted on the transcriptome data, encompassing differential expression gene (DEG) screening, Gene Ontology (GO) functional enrichment, and Kyoto Encyclopedia of Genes and Genomes (KEGG) pathway enrichment. The cucumber genome (http://www.cucurbitgenomics.org, URL (accessed on 6 December 2023)) served as the reference. The DESeq2 method was employed to compare the differences in gene expression between sample groups. The expression level of genes was calculated using Fragments Per Kilobase of transcript per Million mapped reads (FPKM). The criteria for DEGs were set as Fold Change ≥ 1.5 and *p*-value < 0.05.

### 4.5. qRT-PCR

Total RNA extracted by Biomarker Technologies was reverse transcribed into cDNA using the Vazyme Reverse Transcription Kit (R333-01). qRT-PCR was performed using Vazyme’s Fluorescent Quantitative Kit (Q711-02) on a qTOWER3G real-time fluorescent quantitative PCR system, with EF-1α as the internal reference gene. Primers for key DEGs were designed using the NCBI Primer-BLAST tool (https://www.ncbi.nlm.nih.gov/tools/primer-blast/, (accessed on 15 July 2025)) (Primer Premier 6.22) (Table 2). Relative gene expression was calculated using the 2^−ΔΔCt^ method.

### 4.6. Statistical Analysis

Data processing and visualization were conducted using Excel 2010 software. Statistical analysis was performed using SPSS 23.0, with one-way analysis of variance (ANOVA) followed by Duncan’s multiple range test to determine significant differences (*p* < 0.05).

## 5. Conclusions

This study highlights that the foliar application of 150 mg L^−1^ Vc to cucumber seedlings grafted onto pumpkin rootstock using the whip grafting method at the third true leaf expansion stage significantly enhanced cold tolerance during the winter and spring seasons in Hainan. By up-regulating genes involved in photosynthesis, stress signaling, and antioxidant defense, and modulating catalytic activity, binding, transporter activity, and metabolic processes, Vc enhances the antioxidant capacity of grafted seedlings, thereby reducing oxidative damage and improving physiological parameters. These findings provide a theoretical foundation and technical support for optimizing standardized, industrialized, and intensive grafted cucumber seedling production in Hainan, contributing to improving seedling quality and stable yields under cold stress conditions.

## Figures and Tables

**Figure 1 plants-14-02398-f001:**
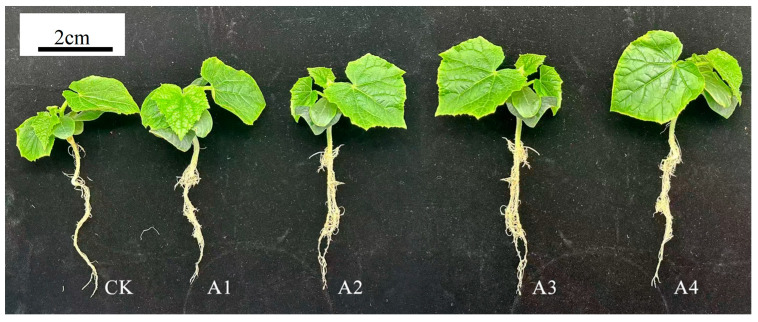
Impact of Vc on growth of grafted cucumber seedlings under low-temperature stress. CK: 0 mg L^−1^, A1: 50 mg L^−1^, A2: 100 mg L^−1^, A3: 150 mg L^−1^, A4: 200 mg L^−1^.

**Figure 2 plants-14-02398-f002:**
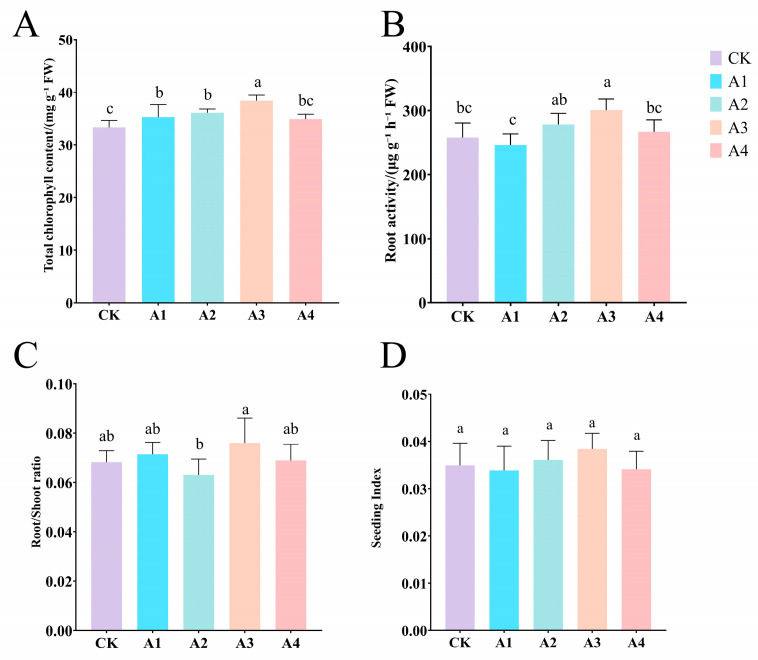
Impact of Vc on physiological indicators of grafted cucumber seedlings under low-temperature stress. (**A**): Total chlorophyll; (**B**): root activity; (**C**): root–shoot ratio; (**D**): seeding index. CK: 0 mg L^−1^, A1: 50 mg L^−1^, A2: 100 mg L^−1^, A3: 150 mg L^−1^, A4: 200 mg L^−1^. In accordance with Duncan’s test, different letters indicate significant differences (*p <* 0.05) among the treatments. The error bars represent mean ± SE (*n* = 3).

**Figure 3 plants-14-02398-f003:**
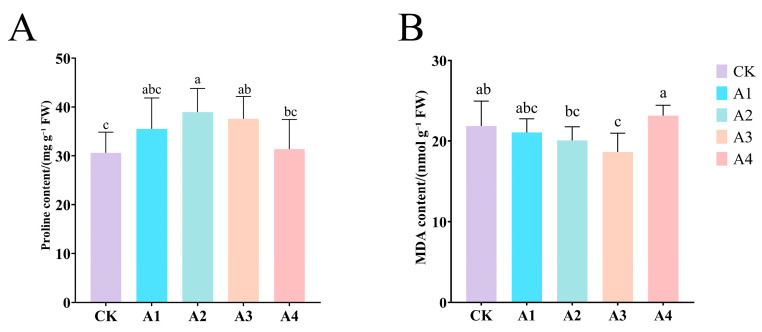
Impact of Vc on permeating substance content of grafted seedlings under low-temperature stress. (**A**): Proline; (**B**): MDA. CK: 0 mg L^−1^, A1: 50 mg L^−1^, A2: 100 mg L^−1^, A3: 150 mg L^−1^, and A4: 200 mg L^−1^. In accordance with Duncan’s test, different letters indicate significant differences (*p* < 0.05) among the treatments. The error bars represent mean ± SE (*n* = 3).

**Figure 4 plants-14-02398-f004:**
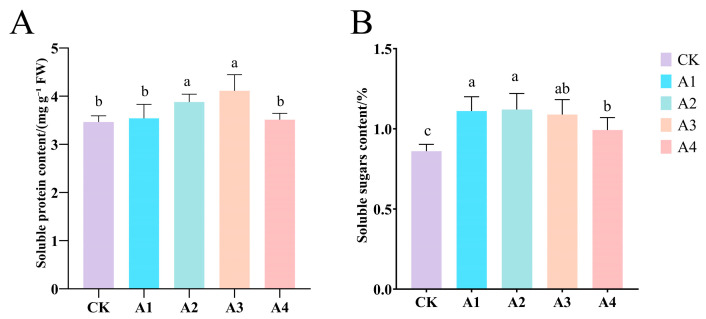
Impact of Vc on the content of osmotic regulatory substances in grafted seedlings under low-temperature stress. (**A**): Soluble protein; (**B**): soluble sugar. CK: 0 mg L^−1^, A1: 50 mg L^−1^, A2: 100 mg L^−1^, A3: 150 mg L^−1^, and A4: 200 mg L^−1^. In accordance with Duncan’s test, different letters indicate significant differences (*p* < 0.05) among the treatments. The error bars represent mean ± SE (*n* = 3).

**Figure 5 plants-14-02398-f005:**
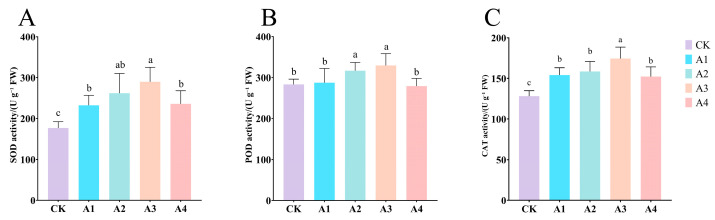
Impact of Vc on antioxidant enzyme activities of grafted seedlings under low-temperature stress. (**A**): Superoxide dismutase; (**B**): peroxidase; (**C**): catalase. CK: 0 mg L^−1^, A1: 50 mg L^−1^, A2: 100 mg L^−1^, A3: 150 mg L^−1^, and A4: 200 mg L^−1^. In accordance with Duncan’s test, different letters indicate significant differences (*p* < 0.05) among the treatments. The error bars represent mean ± SE (*n* = 3).

**Figure 6 plants-14-02398-f006:**
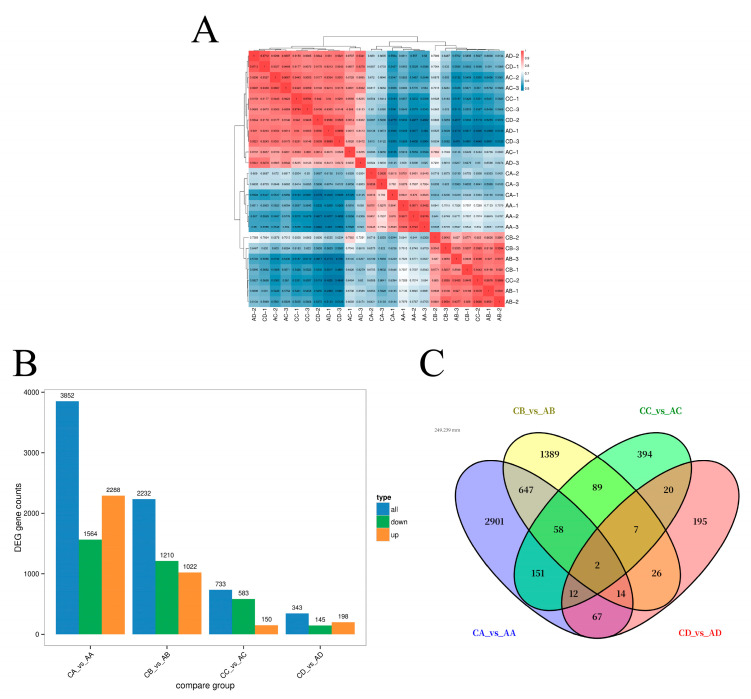
Analysis of transcriptome data at four stages after grafted seedling treatment. (**A**) Clustered heatmap of the expression of all differentially expressed metabolism related genes across the four stages, with red representing up-regulation and blue representing down-regulation. (**B**) Statistics on the number of DEGs at the four stages, where orange represents up-regulated genes, green represents down-regulated genes, and blue represents the total number of DEGs. (**C**) Venn diagrams illustrating the common or unique DEGs within and between control groups at each stage.

**Figure 7 plants-14-02398-f007:**
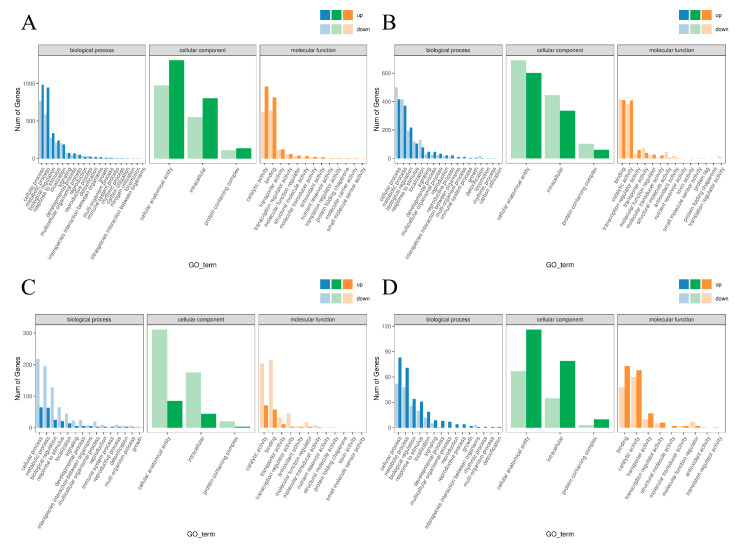
The GO enrichment analysis of DEGs in the comparisons of CA vs. AA (**A**), CB vs. AB (**B**), CC vs. AC (**C**), and CD vs. AD (**D**).

**Figure 8 plants-14-02398-f008:**
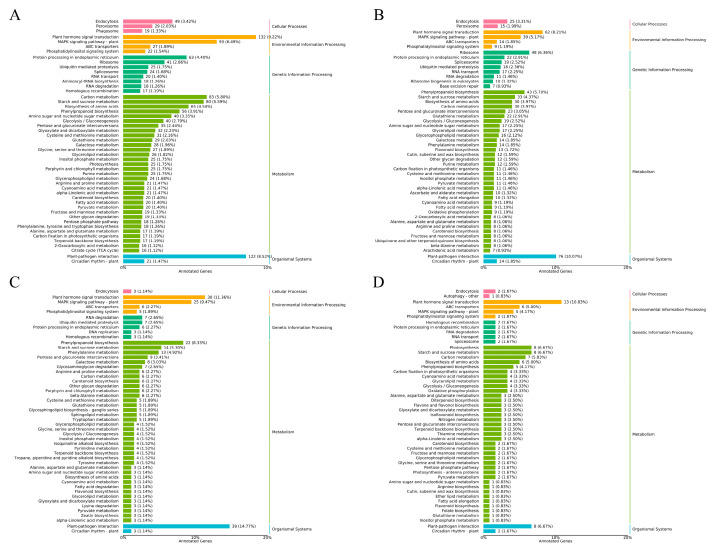
KEGG classification chart of the annotated DEGs in the comparisons of CA vs. AA (**A**), CB vs. AB (**B**), CC vs. AC (**C**), and CD vs. AD (**D**).

**Figure 9 plants-14-02398-f009:**
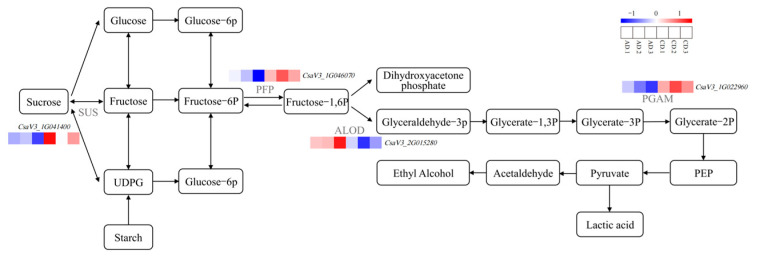
Network diagram of starch and sucrose metabolism and glycolysis/gluconeogenesis pathway. ALOD: fructose-bisphosphate aldolase; SUS: sucrose synthase; PFP: diphosphate-dependent phosphofructokinase; and PGAM: 2,3-bisphosphoglycerate-dependent phosphoglycerate mutase.

**Figure 10 plants-14-02398-f010:**
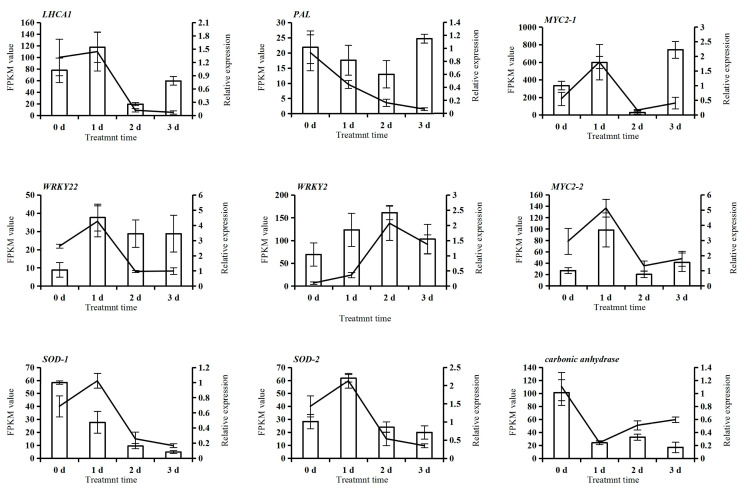
qRT-PCR validation of the selected 9 DEGs. Note: FPKM values for the transcriptome are represented using a line graph, and relative expression for qRT-PCR is represented using a bar graph.

**Table 1 plants-14-02398-t001:** Impact of Vc on the morphological characteristics of grafted seedlings under low-temperature stress.

Treatment	Diameter of Scion Stem (mm)	Height of Scion Plant (cm)	Leaf Area (cm^2^)	Fresh Weight (g)	Chilling Injury Index
CK	3.51 ± 0.11 b	3.42 ± 0.30 b	43.22 ± 3.93 b	3.168 ± 0.371 b	0.580 ± 0.014 a
A1	3.54 ± 0.23 b	3.77 ± 0.23 a	48.86 ± 4.30 b	3.414 ± 0.314 ab	0.552 ± 0.022 ab
A2	3.49 ± 0.25 b	3.70 ± 0.25 ab	64.04 ± 6.31 a	3.538 ± 0.553 ab	0.532 ± 0.020 b
A3	3.84 ± 0.12 a	3.78 ± 0.19 a	58.09 ± 5.76 a	3.787 ± 0.364 a	0.518 ± 0.015 b
A4	3.44 ± 0.20 b	3.52 ± 0.27 ab	48.57 ± 6.28 b	3.068 ± 0.483 b	0.536 ± 0.020 b

Note: CK: 0 mg L^−1^, A1: 50 mg L^−1^, A2: 100 mg L^−1^, A3: 150 mg L^−1^, A4: 200 mg L^−1^. Error bars represent 1 SE (*n* = 3) within the same column, and different letters indicate a significant difference between treatments (*p* < 0.05).

**Table 2 plants-14-02398-t002:** Primer sequences used for qRT-PCR in this study.

Gene Name	Gene ID	Forward Primers (5′–3′)	Reverse Primers (5′–3′)
*EF-1α*	*EF-1α*	ACTGGTGGTTTTGAGGCTGGT	CTTGGAGTATTTGGGTGTGGT
*PAL*	*CsaV3_6G014060*	TCGTCCTAATGCCAAGGCTG	ACCTCGGACAACAAAGCCAA
*MYC2-1*	*CsaV3_3G001710*	GCGATTTTCTGGCAGTCGTC	CGTAACCTCCTCATCCACCG
*MYC2-2*	*CsaV3_3G007980*	CAGAGTGAGGTCGTCAGCAG	TGGCACATGCTTCCCATGAT
*WRKY22*	*CsaV3_4G034570*	GGATATCAAGCCGTCGCAGA	CGGAGGTACCGAAATCGGAG
*WRKY2*	*CsaV3_3G026920*	GGAGGCGGTAAGTGGGTTTT	AGCTCCTCGGGAACTTGTTG
*SOD-1*	*CsaV3_4G013220*	TGCTCTTGGCGATACCACAA	ACGACTGCCCGACCAATTAC
*SOD-2*	*CsaV3_6G014400*	CGCTGTCCTCAAGGGAACTT	ATGTCGGATTTCGTCCTCGG
*carbonic anhydrase*	*CsaV3_2G030320*	AGCAAACATTGTTCCGCCAT	AGCTGCTTTTGCATTCACCA
*LHCA1*	*CsaV3_5G025740*	CCATGTCCGCTGAGTGGATG	GGAACTGCTGCCCATTCTTG

## Data Availability

All data generated or analyzed during this study are included in this published article.

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
