# Peer review of "Physiological and Transcriptomic Mechanisms Underlying Vitamin C-Mediated Cold Stress Tolerance in Grafted Cucumber"

_plants, 2025, doi:10.3390/plants14152398_

Round 1
Reviewer 1 Report
Comments and Suggestions for Authors
The authors prepared the manuscript on Physiological and Transcriptomic Mechanisms Underlying Vitamin C-Mediated Cold Stress Tolerance in Grafted Cucumber. This study investigates the physiological and molecular mechanisms by which exogenous vitamin C (Vc) mitigates cold stress in grafted cucumber seedlings.
Organize your citation. Style.
When describing the results, do not just provide information about what is the most, what is the least (we can see this from tables and figures). Provide how strong the effect of the treatment was, etc.
Since the methodology is at the end of the manuscript, explanations of processing abbreviations should be provided next to the tables and figures.
In figure 6, the numbers are blurred, so even when enlarged, it is difficult to see, a better quality image is needed.
Figure 7,8 the text is also blurred.
Reviewer 2 Report
Comments and Suggestions for Authors
The authors investigated the effects of foliar application of vitamins on cucumber plants and, through integrated physiological and molecular analyses, demonstrated that vitamin C treatment significantly enhances cold stress tolerance in cucumber. However, it would be beneficial if the authors first present data on how vitamin C application affects cucumber plants under normal (non-stress) conditions. Specifically, Figure 1 could be utilized to show the physiological or molecular responses of cucumber to vitamin C treatment under standard conditions, thereby providing a baseline for understanding the mechanism of action. This approach would help to establish the mode of vitamin C uptake and its initial impact before assessing its role in enhancing tolerance under cold stress conditions.
It is necessary to include a fundamental experimental design that demonstrates how the mode of vitamin C application influences its effectiveness. Such a foundation would provide context for the subsequent analysis under cold stress conditions. Additionally, it may be more appropriate to combine the data currently presented in Figures 1 and 2 into a single figure, as separating them does not appear essential.
While the statistical analyses are appreciated, it would also be helpful to include representative images of whole plants and grafted seedlings to support the quantitative data visually. Moreover, the number of biological replicates (sample size) should be clearly indicated in each graph. Ultimately, Figures 1 through 3 could be merged into a comprehensive figure, which would improve clarity and organization. For Figure 3 in particular, the inclusion of sample size information and a positive control would strengthen the interpretation.
Regarding Figure 9, the rationale connecting the experimental data to the conclusions drawn is not clearly established. If the biological pathway illustrated in Figure 9 is proposed to result from vitamin C-induced changes in gene expression or metabolism in cucumber, this connection should be explicitly supported by the data. In that case, the qRT-PCR analysis shown in Figure 10 should ideally validate genes that are directly related to the pathways or processes hypothesized in Figure 9. However, Figure 10 appears to present genes selected primarily based on high expression levels observed in the RNA-seq dataset, rather than on their relevance to the proposed mechanism.
This does not mean that the overall research approach is fundamentally flawed. However, it raises a broader question regarding the practical implications of applying exogenous vitamin C in crop production. Given that vitamin C is typically considered an antioxidant and nutritional supplement for human consumption, the authors may want to further clarify the objective and potential agricultural relevance of foliar vitamin C application in cucumber.
Round 2
Reviewer 1 Report
Comments and Suggestions for Authors
The authors made important corrections, taking into account the reviewer's comments. After completing the proper formatting, the quality of the manuscript is quite good.